# Position: AI Capabilities May Not Increase Exponentially

Haosen Ge[1]    Hamsa Bastani[2]    Osbert Bastani[3]

## Abstract

Rapidly increasing AI capabilities have substantial real-world consequences, ranging from AI safety concerns to labor market consequences. The Model Evaluation & Threat Research (METR) report argues that AI capabilities have exhibited exponential growth since 2019. In this position paper, we argue that the data is insufficient to support exponential increase in AI capabilities. We propose an alternative hypothesis that existing exponential growth has been driven by (1) model and data scaling, and (2) the development of reasoning models. Barring major new breakthroughs, our hypothesis is that AI capabilities have already exhibited an inflection point or will do so in the near future. We call for more rigorous evaluation methodologies for AI forecasts and better academic discussion on this topic.[1]

## 1. Introduction

Pretrained large language models (LLMs) have demonstrated that broad AI capabilities can arise largely from a combination of large-scale pretraining (Brown et al., 2020) and chain-of-thought reasoning (Wei et al., 2022; Yao et al., 2023). This fact points to a natural path to achieving ever-stronger AI capabilities: simply scale pretraining to larger models and datasets while incorporating post-training to improve reasoning. This recipe has successfully driven rapid improvements in AI capabilities over the past few years.

The rate of progress has set expectations that LLMs may

---

[1]Wharton AI & Analytics Initiative, The Wharton School, University of Pennsylvania, USA [2]Department of Operations, Information and Decisions, The Wharton School, University of Pennsylvania, USA [3]Department of Computer and Information Science, University of Pennsylvania, USA. Correspondence to: Haosen Ge <hge@wharton.upenn.edu>, Hamsa Bastani <hamsab@wharton.upenn.edu>, Osbert Bastani <obastani@seas.upenn.edu>.

*Proceedings of the $43^{rd}$ International Conference on Machine Learning*, Seoul, South Korea. PMLR 306, 2026. Copyright 2026 by the author(s).

[1]Our code is available at: https://github.com/METR/eval-analysis-public..

---

achieve broad expert-level capabilities in the next few years. A recent report by Model Evaluation & Threat Research (METR) (Kwa et al., 2025) conducted a series of analyses to measure AI capabilities in realistic tasks that require significant effort from human experts. They propose a novel metric: 50% model horizon, which measures the difficulty of tasks that a model can solve successfully 50% of the time. Then, they show that according to this metric, AI capabilities are increasing exponentially over time—specifically, model horizons have been doubling every seven months since 2019. Based on these results, they predict that "within 5 years, AI systems will be capable of automating many software tasks that currently take humans a month."

Such predictions have raised substantial alarms about the dangers of advanced AI capabilities. Much of the early conversation revolved around concerns about AI safety risks (Amodei et al., 2016; Barnett & Scher, 2025). However, there are substantial consequences for rapidly increasing AI capabilities beyond safety. Most notably, these results have raised labor-market concerns about the potential for large-scale displacement of skilled workers (Eloundou et al., 2023; Brynjolfsson et al., 2025). Many consequences of these forecasts are immediate, shaping both policy outcomes as well as individual decisions such as choices about education and career paths. Given the substantial consequences of the potential for exponential increase in AI capabilities, there is an urgent need for rigorous methodologies for performing and validating these kinds forecasts.

**In this position paper, we argue that exponential growth may not continue for much longer.** While there has been heated debate about the plausibility that growth will soon slow, little rigorous evidence has been put forth to support this idea. In general, it is impossible to determine whether growth will continue exponentially or plateau based on historical data alone. However, we argue that we can make plausible predictions about plateauing growth on the basis of domain knowledge about how LLMs have improved over time. That is, rather than simply extrapolating historical trends, we must analyze the mechanisms that drove those trends and make informed hypotheses about how these mechanisms may evolve in the future.

To demonstrate this approach, we posit a theoretical model under which the exponential appearance of recent gains

in AI capabilities can be interpreted as a consequence of the introduction of reasoning capabilities into base LLMs. Specifically, we model reasoning as a separate technology that contributes multiplicatively to the overall capability of LLMs—i.e., LLM capabilities can be decomposed into two sigmoids, one for the base LLM and one for reasoning. Then, our hypothesis is that following initial exponential growth due to scaling data and model size, base capabilities plateaued, but overall capabilities continued to grow for a period due to rapid improvements in reasoning. Under our model, innovation will plateau unless there are substantial new innovations that continue to drive exponential progress.

While we present a specific alternative analysis, our goal is not to discount the METR study; in fact, we believe continuing exponential improvement is a plausible viewpoint, and it is important to take this potential outcome into consideration. Instead, we argue that it is critical for the academic community to perform rigorous work on forecasting methodologies, and highlight opportunities for doing so. We present a "call to action" highlighting a potential path towards more rigorous forecasting: (1) developing rigorous methodologies for designing and evaluating forecasts, and (2) establishing venues for academic discussion of forecasting work.

**Discussion of recent progress.** Since the initial posting of this paper, several models have been released and added to the METR forecasts. While these data are not incorporated directly into our analysis, we discuss them briefly here. The series of models released by OpenAI would appear to support our hypothesis—according to METR's analysis, their capabilities peaked with GPT-5.2 and have remained flat with GPT-5.3-Codex and GPT-5.4 (the results for GPT-5.5 are still pending at this time). The models released by Anthropic paint a more complicated picture, with a rapid increase in capabilities with Claude Opus 4.6 (with results for Opus 4.7 still pending). Furthermore, the release of Claude Mythos—potentially the first successful result in model scaling since GPT-4—is a significant milestone in AI capabilities. It remains to be seen whether this breakthrough continues to drive progress; however, we believe it is unlikely as another step of model scaling would require multiple tens of trillions of parameters. Broadly, we believe the question of whether AI capabilities have already hit an inflection point or will do so soon remains open.

## 2. Alternative Views

**AI capabilities are increasing exponentially.** This is the main alternative view. The METR study acknowledges that it is impossible to guarantee that progress will not plateau. Furthermore, in Appendix D.1, they claim that they compare to alternatives including a sigmoid curve, but find that they do not find "any evidence that AI horizon is leveling off (on our metric)" according to this methodology. However, we find that even using a direct strategy where we fit a sigmoid curve to their data, we find that the inflection point of the sigmoid has already passed (specifically, 2025-06-06). It appears plausible that progress since the date of the release of o1-preview (specifically, 2024-09-12) is linear rather than exponential, which is consistent with the fact that the linear region of the sigmoid curve spans the period between that date and the present. While we cannot definitively rule out exponential increase, we believe that alternatives such as plateauing growth are at least substantially more plausible than the METR study would suggest.

**Forecasting AI capabilities is unnecessary.** Another alternative view is that there is little value in producing accurate forecasts. This was traditionally true: even though machine learning has progressed rapidly in the past decade, the speed of progress has not had substantial practical impact. Note that we distinguish the capabilities and properties of *current models* (which have long been relevant) from projections about the capabilities of *future models*. Specifically, the projected capabilities of AI models in a few years is already influencing decisions about education and labor policy, both at an individual and societal level. For instance, undergraduate students may avoid a certain major based on the assumption that the jobs deriving from that major will be obsolete by the time they graduate due to AI automation.

**Forecasting AI capabilities is best left to others.** Another view is that forecasting AI capabilities is not an academic endeavor, or not one that should be undertaken by the machine learning community. Indeed, this kind of work is not typical of the machine learning community, whose main goal has been to create more capable artificial intelligence. However, we argue that machine learning researchers can make substantial contributions to AI forecasts. Unlike forecasts based purely on historical trends, we propose that forecasting should use domain knowledge to decompose progress into multiple components, understand the rates at which separate components are progressing, and then combine these component-specific forecasts back into a forecast of overall capabilities. This kind of approach requires substantial domain knowledge about how AI models are designed and trained. As an example, the machine learning community has already started studying scaling laws, which could help form the basis of more accurate forecasts.

**Forecasting AI capabilities is too difficult.** Finally, another view is that forecasting AI capabilities is simply too difficult since there is so little data available, and it furthermore requires extrapolating into the future based on unknown factors. We argue that forecasts should be grounded in domain knowledge, accounting for how models are improving along various components (e.g., base vs. reasoning capabilities, but also pre- and post-training techniques, data engineering, architectural improvements etc.) rather than purely based on

*Table 1.* Release Dates of the Selected SOTA Models.

| Model | Release Date |
|---|---|
| GPT-4 | 2023-03-14 |
| GPT-4 (1106) | 2023-11-06 |
| GPT-4o | 2024-05-13 |
| GPT-o1-preview | 2024-09-12 |
| Gemini-3 Pro | 2025-11-18 |
| Claude Opus 4.5 | 2025-11-24 |
| GPT-5.2 | 2025-12-11 |
| GPT-o1-Inspect | 2024-12-05 |
| Claude 3.7 Sonnet | 2025-02-24 |
| Claude 3.5 Sonnet | 2024-06-20 |
| GPT-5 | 2025-08-07 |

curve fitting at the aggregate level. However, we acknowledge that forecasting is by its nature a challenging problem, and no single forecast can "solve" the problem. Thus, we also argue that a diverse range of techniques must be developed to address the problem, and the resulting variation in forecasts can be accounted for in decision-making.

## 3. Background on the METR Study

We focus on the recent METR study (Kwa et al., 2025), which forecasts that AI capabilities are exponentially increasing. This study was itself critiquing prior work, pointing out that existing metrics such as accuracy are bounded and cannot assess whether growth is exponential. To remedy this issue, they introduce a novel metric, the *50% model horizon time*, which quantifies the difficulty of tasks that a model can solve reliably. Unlike prior metrics, this one can increase unboundedly, making it suitable for assessing the possibility of exponential growth. Their analysis concludes that AI capabilities are improving exponentially.

Their experiments include three task families: HCAST, RE-Bench, and SWAA. Specifically, HCAST contains a diverse set of challenges in cybersecurity, machine learning, software engineering, and general reasoning. RE-Bench consists of challenging open-ended machine learning research engineering environments, each of which are intended to take a human expert approximately 8 hours to complete. SWAA comprises 66 small tasks commonly performed in software engineering work. In total, their study includes 170 unique tasks from the three task families. Then, they evaluate 28 popular models on the 170 tasks. Among the models, they label a subset of 15 models as state-of-the-art (SOTA), representing the frontier of AI capabilities. We reuse their experimental results and focus exclusively on the 15 state-of-the-art models to better characterize the scaling behavior of frontier AI capabilities. We include the list of SOTA models and their release dates in Table 1.

Their data analysis first estimates the horizon time of each model on each dataset using the following regression:

$$p_{\text{model}} = \sigma((\log h_{\text{model}} - \log t_{\text{task}}) \cdot \beta_{\text{model}}), \qquad (1)$$

where $p_{\text{model}}$ denotes the probability that the model solves a task correctly, $t_{\text{task}}$ denotes the difficulty of the task (measured by the amount of time human expert takes to complete the task), $h_{\text{model}}$ is the 50% horizon time, $\beta_{\text{model}}$ is the parameter they estimate, and $\sigma(x) = e^x/(1 + e^x)$ is the sigmoid function. By construction, $p_{\text{model}} = 1/2$ when $\log h_{\text{model}} = \log t_{\text{task}}$; thus, a model with capability $h_{\text{model}}$ attains a success probability of $1/2$ on tasks with difficulty $t_{\text{task}} = h_{\text{model}}$. Thus, $h_{\text{model}}$ characterizes the task difficulty threshold at which the model achieves a success rate of $1/2$.

After estimating the 50% horizon time, they examine the temporal trend in model capabilities by fitting the following linear regression model:

$$\log h_{\text{model}} = \beta_0 + \beta_1 \cdot d_{\text{model}},$$

where $d_{\text{model}}$ denotes the model's release date; they fit this data by treating the estimate $h_{\text{model}}$ from the previous step as ground truth $h_{\text{model}}^*$ and applying linear regression. Note that this model is equivalent to

$$h_{\text{model}} = \exp\left(\beta_0 + \beta_1 \cdot d_{\text{model}}\right), \qquad (2)$$

implying an exponential relationship between the model's $50\%$ horizon time and its release date. METR reports an $R^2$ value of 0.98 for this regression, which they interpret as strong evidence that the capabilities of frontier models grow exponentially over time. In their main paper, they compare this regression to two others—linear and hyperbolic—and find that an exponential curve fits the data substantially better. They discuss additional comparisons informally in Appendix D.1 of their paper, but do not provide quantitative evidence to support ruling out these alternatives. However, this limited comparison to alternatives makes it difficult to assess the confidence we should have in their findings.

## 4. Stacked Model of AI Progress

We present a model of growing AI capabilities that decomposes it into two component technologies: the base model and reasoning; our model can easily be extended to incorporate additional components if substantial new breakthroughs occur. We prove that this model produces a growth curve that is qualitatively consistent with the METR data.

### 4.1. Motivation

The main hypothesis behind our model of AI progress is that much of the recent growth in AI capabilities has been driven by the introduction of reasoning into base LLMs. Specifically, while chain-of-thought reasoning has been popular

for some time now (Wei et al., 2022), explicitly training models to perform reasoning is a more recent phenomenon, starting with OpenAI's o1 model (Jaech et al., 2024; Shao et al., 2024). Since the release of o1, there has been startling progress on a number of benchmarks, with current LLMs appearing to approach the performance of human experts.

We propose a model that explicitly separates progress on base model capabilities from progress on reasoning capabilities. Intuitively, the reason this model might show support for slowing improvements is that technologies tend to exhibit very rapid growth during their introduction. Since LLMs have only been finetuned for chain-of-thought reasoning for a little over a year (starting with the introduction of OpenAI's o1 model (Jaech et al., 2024)), it is natural that reasoning capabilities have dramatically improved capabilities over the past year. However, we might expect reasoning capabilities to start plateauing in the near future.

Without separating out improvements in reasoning, it is unsurprising that the rate of improvement appears exponential—prior to 2023, base capabilities improved exponentially due to scaling of data and model size, but these improvements plateaued due to the prohibitive cost of scaling. Reasoning capabilities have maintained this growth since 2023. Under this view, one way to interpret the METR study is that it predicts new breakthroughs will continue to prop up exponential progress. This makes sense when extrapolating, since breakthroughs have been common in the past decade; however, there is no guarantee that it will continue to be the case. Under our model, if breakthroughs stop happening, then exponential progress will end.

## 4.2. Regression Model

For our data analysis, we consider an alternative regression model to Eq. 2 that explicitly separates a model's base performance and its reasoning capabilities:

$$h_{\text{model}} = \gamma_1 \cdot h_{\text{base}} \cdot (1 + \gamma_2 \cdot h_{\text{reasoning}})$$
$$h_{\text{base}} = f(d_{\text{model}})$$
$$h_{\text{reasoning}} = g(d_{\text{model}}) \cdot 1\{k_{\text{thinking}} = 1\},$$

where $h_{\text{model}}$ is the 50% model horizon, $k_{\text{thinking}}$ indicates whether the model has been post-trained with reasoning capabilities and those capabilities are activated, $\gamma_1, \gamma_2$ are parameters to be estimated, and $f, g$ are link functions described below. In other words, we treat an LLM's overall capability as the product of it's base capability $h_{\text{base}}(d)$ (i.e., without reasoning) and the quality $h_{\text{reasoning}}$ of its reasoning features. Intuitively, the base capability of a LLM captures advancement of the model's pre-training phase and non-reasoning post-training phases, including model sizes and data curation, and the reasoning ability captures post-training techniques to improve chain-of-thought thinking. Both base and reasoning capabilities are functions of

the model's release date $d_{\text{model}}$. The parameters $\gamma_1$ and $\gamma_2$ quantify the contributions of base capability and reasoning capability to the overall capability.

A key feature of our model is that technologies are *multiplicative*—i.e., overall capability is the product (rather than, e.g., the sum) of the component technologies (in our case, the base model and reasoning). This assumption is the key driver behind the apparent exponential growth of staggered improvements across different technologies; we provide theoretical evidence that our model exhibits this kind of behavior in Section 4.4. Intuitively, just as progress in one technology plateaus, another technology exhibits rapid improvement that props up exponential improvements. We believe this stacked model is realistic—for LLMs, reasoning cannot exist without strong base models, and while strong base models have useful capabilities, these are substantially boosted by reasoning.

## 4.3. Choice of Link Functions

The link functions $f, g$ encode how the base and reasoning capabilities depend on the model's release date $d_{\text{model}}$. A wide range of functional forms could be considered; in this paper, we adopt the following plausible candidates.

**Sigmoid.** First, we consider the sigmoid function:

$$f(d_{\text{model}}) = \sigma(\delta_1 \cdot d_{\text{model}} + \delta_2) \tag{3}$$
$$g(d_{\text{model}}) = \sigma(\theta_1 \cdot d_{\text{model}} + \theta_2), \tag{4}$$

where $\delta_1, \delta_2, \theta_1, \theta_2$ are parameters. Intuitively, these link functions say that each of base and reasoning capabilities grow exponentially until reaching an "inflection point", after which they plateau (formally, an inflection point of an arbitrary function $f(x)$ is the point at which $f''(x)$ changes sign; the inflection of $\sigma$ is at $x = 0$). In Section 4.4, we provide an analysis of the implications of this model.

**Exponential.** Next, we consider an exponential function:

$$f(d_{\text{model}}) = \exp(\delta_1 \cdot d_{\text{model}} + \delta_2)$$
$$g(d_{\text{model}}) = \exp(\theta_1 \cdot d_{\text{model}} + \theta_2),$$

where $\delta_1, \delta_2, \theta_1, \theta_2$ are parameters. Similar to METR's original model, these link functions say that each of base and reasoning capabilities are increasing exponentially in time; thus, the overall capability is also increasing over time.

**Spline.** Lastly, we consider a B-spline link function:

$$f(d_{\text{model}}) = \sum_{i=1}^{N_f} \delta_i \cdot B_i(d_{\text{model}})$$
$$g(d_{\text{model}}) = \sum_{i=1}^{N_g} \theta_i \cdot B_i(d_{\text{model}}),$$

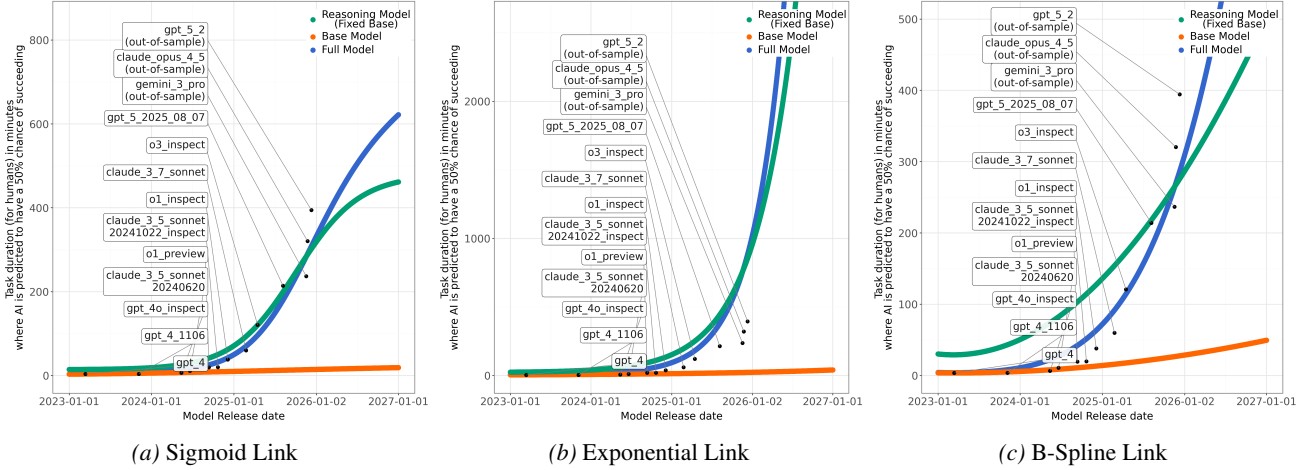

*(a)* Sigmoid Link      *(b)* Exponential Link      *(c)* B-Spline Link

*Figure 1.* **Projections under Different Link Functions.** The orange curves project base model capabilities, the green curve projects reasoning capabilities assuming the best base model (i.e., gpt-5.1-codex-max), and the blue curve shows the overall capabilities. The black points denote the 50% model horizon estimated by METR.

where $N_f$ and $N_g$ denote the numbers of spline basis functions for $f$ and $g$, respectively, $\{\delta_i\}_{i=1}^{N_f}$, $\{\theta_i\}_{i=1}^{N_g}$ are parameters, and $B_i(d_{\text{model}})$ is the $i$th B-spline basis function (which we take to be a degree $m$ polynomial), which includes its own parameters. This choice of link function induces a flexible, piecewise polynomial relationship between LLM capabilities and the release date.

### 4.4. Theoretical Analysis

We prove that a stacked model of technological progress with sigmoid link exhibits exponential growth followed by plateauing. To simplify our analysis, we drop many of the parameters and assume that the core model is a product of sigmoid functions with different inflection points; we further simplify by assuming these inflection points are evenly spaced. Then, we prove that the resulting function exhibits (i) exponential growth $e^x$ before the first inflection point, (ii) squared exponential growth $e^{x^2}$ between the first and last inflection points, and (iii) plateauing thereafter.

**Theorem 4.1.** *Let $x$ denote time, and consider the model*

$$f(x) = \prod_{i=1}^{k} f_i(x) \quad where \quad f_i(x) = \sigma(x - i\alpha),$$

*where $\sigma$ is the sigmoid function and $\alpha \geq 2$. Then:*

- *If $x \leq 0$, then*

$$\frac{1}{5} e^{kx} \exp\left(-\frac{\alpha}{2} \cdot k(k+1)\right)$$
$$\leq f(x) \leq e^{kx} \exp\left(-\frac{\alpha}{2} \cdot k(k+1)\right).$$

- *If $x \in [j\alpha, (j+1)\alpha]$ for $j \in \{0, 1, ..., k-1\}$, then*

$$\frac{1}{20} \exp\left(-\frac{\alpha}{2} \cdot (k-j+1)(k-j)\right)$$
$$\leq f(x) \leq \exp\left(-\frac{\alpha}{2} \cdot (k-j-1)(k-j)\right).$$

- *If $x \geq k\alpha$, then*

$$\frac{1}{4} \leq f(x) \leq 1.$$

In the first case $x \leq 0$, the model exhibits exponential growth $e^{kx}$. In the second case, $x \approx j\alpha \approx (j+1)\alpha$, so

$$f(x) \approx \exp\left(-\frac{\alpha}{2} \cdot \left(k - \frac{x}{\alpha}\right)^2\right).$$

In other words, progress continues exponentially until the final inflection point $k\alpha$. Intuitively, the exponent base decays throughout this phase; as progress for individual components plateau, the base of the exponent becomes smaller. Finally, in the third case, progress plateaus once the last inflection point has been crossed.

These trends are consistent with our empirical analysis in Section 5. At a high level, the initial period of growth was exponential due to scaling AI capabilities. Many researchers believed that capabilities were plateauing; however, the introduction of post-training for reasoning sparked a second wave of improvements in capabilities, resulting in a period of steep linear increases between 2024-09-12 to the present. If our model accurately reflects reality, then new breakthroughs are necessary to sustain exponential growth.

*Table 2.* Assessment of goodness of fit via MSE on $h_{\text{model}}$.

| Specification | Mean Squared Error (MSE) |
|---|---|
| Sigmoid Link | 274.20 |
| B-Spline Link | 651.67 |
| Exponential Link | 23609.29 |
| METR Exponential Curve | 1095.16 |

## 5. Our Analysis of the METR Data

We fit our model from Section 4 to data; our results support the plausibility of plateauing AI capabilities.

### 5.1. Methodology

We estimate our models the data shared by the METR study (Kwa et al., 2025) (including data from the HCAST, RE-Bench, and SWAA benchmarks). We access the data via METR's public Github repository: `https://github.com/METR/eval-analysis-public`; we use the "Time Horizon 1.1" version.

We perform estimation by maximizing the log-likelihood of the probabilistic model $p_{\text{model}}$ in Eq. 1 using Stan (Carpenter et al., 2017). For the sigmoid and exponential link functions, we adopt the weakly informative prior $\mathcal{N}(0, 10^2)$ for all parameters, and impose positivity constraints on $\gamma_1, \gamma_2, \delta_1, \theta_1, \beta_{\text{model}}$.

For the B-spline link function, we use two breakpoints and polynomial splines of degree two for both $f$ and $g$ (i.e., degree $m = 2$ and $N_f = N_g = 2 + 2 - 1 = 3$).

To ensure that $h_{\text{model}}$ remains strictly positive and to avoid taking the logarithm of a negative quantity, we constrain all spline coefficients $\delta_i$ and $\theta_i$ to be positive. Furthermore, we regularize the spline coefficients using random-walk priors, following standard practice to mitigate overfitting: $\delta_1 \sim \mathcal{N}(0, 1)$, $\delta_i \sim \mathcal{N}(\delta_{i-1}, \tau)$ for all $i \geq 2$, and $\tau \sim \mathcal{N}(0, 1)$. The same priors are applied to $\theta_i$.

### 5.2. Results

**Model Fit.** Figure 1 shows the fitted models; that combine base and reasoning components. Given the limited number of available models, we assess goodness-of-fit using the in-sample mean squared error (MSE) between the model-predicted AI capability and the observed AI capability: $(h_{\text{model}} - h^*_{\text{model}})^2$; Table 2 reports the resulting MSEs. We also include METR's exponential curve and a sigmoid curve;[2] note that these models are not directly compara-

---

[2]This sigmoid curve is fit by minimizing the mean-squared error (MSE) of the curve $h_{\text{model}} = \gamma \cdot \sigma(\delta_1 \cdot d_{\text{model}} + \delta_2)$ to the METR dataset, where $h_{\text{model}}$ is METR's "50% model horizon time" for the given model, $d_{\text{model}}$ is the model release date, and $\gamma, \delta_1, \delta_2$

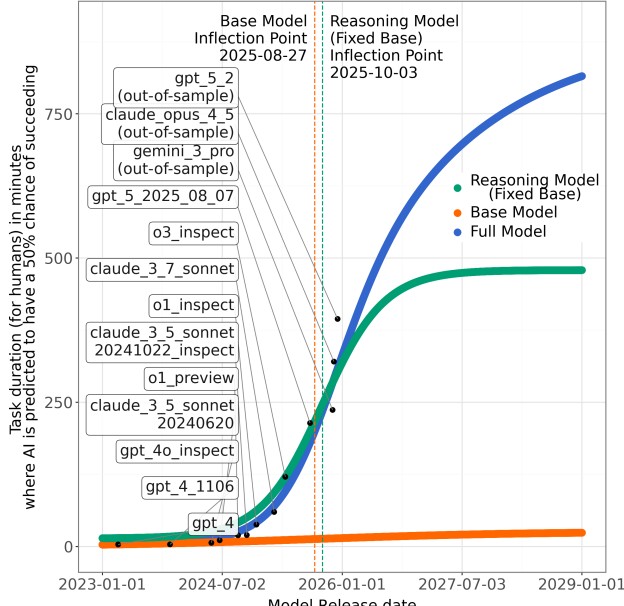

*Figure 2.* **Sigmoid link inflection points.** The curves are as in Figure 1; we show inflection points of the orange and green curves as dashed vertical lines of the corresponding color.

ble since they optimize different loss functions. Among the three link functions considered, the sigmoid-link model achieves by far the lowest MSE, suggesting that within our model, sigmoid growth appears more plausible than exponential growth. Our approach also outperforms METR's exponential curve, though our model has more parameters so this comparison is not rigorous.

**Inflection points.** The key question is not the magnitude of recent performance gains (which are undeniable), but whether these gains will continue. One argument in the METR study is that there is no sign of an "inflection point" where exponential increase starts to slow down. To further understand the inflection points in AI capabilities, we plot the inflection points of Eqs. (3) & (4) in Figure 2. The inflection points $t_f$ and $t_g$ of $f(t)$ and $g(t)$, respectively, based on the estimated parameters are

$$\hat{t}_f = 2025\text{-}08\text{-}27 \qquad \text{and} \qquad \hat{t}_g = 2025\text{-}10\text{-}03,$$

respectively. The inflection points for both base model and reasoning capabilities have recently passed, suggesting that new breakthroughs may be needed to continue to drive progress (though inflection point estimates may be noisy).

Baked into our model is the idea that unless significant breakthroughs happen, progress will plateau. Thus, the question about forecasting AI capabilities becomes a question of whether we should expect another breakthrough that produces improvements on the same scale as reasoning.

---

are parameters. We use gradient descent for parameter estimation.

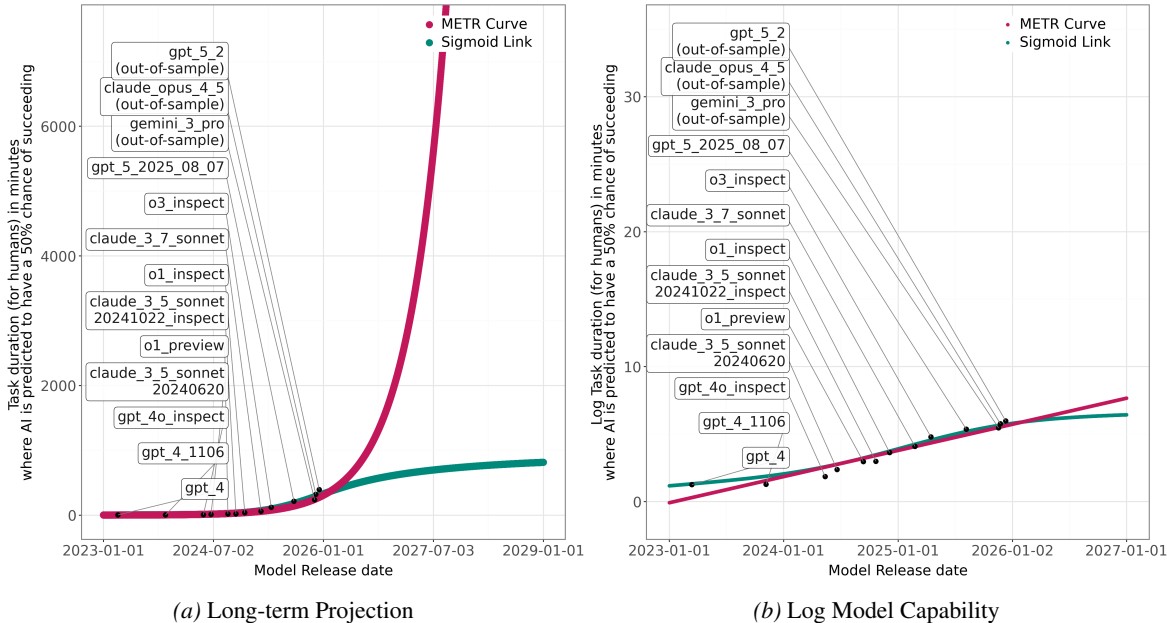

*(a)* Long-term Projection    *(b)* Log Model Capability

*Figure 3.* **Comparison of Sigmoid Link and METR Projection.**

This question can only be answered by domain knowledge; we leave it for the broader community to answer.

These results highlight an additional benefit of decomposing overall capability into component technologies—it enables us to understand and interpret forecasts of progress for individual components separately, providing an understanding of *why* progress looks a certain way.

**Long-term forecasts.** Finally, we compare METR's exponential forecast with our sigmoid-link model over a six-year horizon from 2023-01-01 to 2029-01-01. Results are shown in Figure 3a. The two models yield similar projections up to approximately 2026-03-01. Beyond this point, the METR model predicts an increasingly rapid rise in model capability, whereas our model suggests that capabilities will plateau in the near future. External forecasts cannot in general be validated, so these results highlight the importance of leveraging domain insights to assess the validity of forecasts.

## 6. Limitations

**In-sample evaluation.** A key limitation is that our estimates are all evaluated in-sample; while the same is true for existing methodologies such as the METR study, our models also include more parameters. This limitation is inevitable due to the limited amount of data available. However, our goal is not to provide irrefutable evidence that AI capabilities are plateauing, but that it is a highly plausible alternative to continuing exponential growth. As more data becomes available, it is critical to assess which models have more accurately forecast progress to build confidence in future

forecasts. More broadly, more rigorous methodologies must be developed for assessing the accuracy of these forecasts.

**Evaluation metric.** A related issue is that comparing across different kinds of models (ours vs. METR) is complicated by the fact that they are estimated in very different ways. The METR paper performs regression to minimize the MSE in the space of log-outcomes $\log h_{\mathrm{model}}$, whereas we have used probabilistic modeling directly on the final outcomes $p_{\mathrm{model}}$. While we have used MSE at predicting $h_{\mathrm{model}}$ to compare models, this comparison is not fair due to the diverse loss functions, especially given the in-sample comparison. Again, this issue necessitates the development of more rigorous evaluation methodologies.

**Multiplicative assumption.** The multiplicative assumption is critical for driving our theoretical analysis. While we believe this assumption presents at least a plausible alternative, substantial work needs to be done to validate it in practice.

**Limited decomposition.** We have only modeled base and reasoning capabilities; we believe these are the key driving factors behind the substantial recent gains in AI capabilities, but there are many other important factors. Prior to the release of GPT-4, many of the gains in performance were driven simply by scaling the amount of training data and the model size, as well as basic instruction tuning. Recent improvements in reasoning capabilities have been driven by improved post-training procedures and the creation of datasets tailored to reasoning tasks; other non-public improvements may also have contributed. In principle, these two components can be further decomposed into aspects such as data engineering, pre- and post-training algorithms, network

architecture, etc.; however, we currently lack enough information about state-of-the-art LLMs to perform a more granular analysis. Finding ways to address these issues could help lead to more accurate forecasts.

## 7. Call to Action

Our analysis is not intended to be a definitive forecast of AI capabilities, nor even a rigorous one. Instead, our main goal is to highlight that existing forecasts of exponential increase are insufficiently supported by the data, and that plateauing growth is a highly compelling alternative.

**More rigorous forecasting methodologies.** Forecasting is intrinsically an extrapolation problem, which is in general unsolvable. Existing forecasting methodologies are largely based on naïvely extrapolating from historical data. We argue that more rigorous forecasts require incorporation of domain-specific insights. We have presented one possibility—namely, decomposing LLM capabilities into component technologies, forecasting progress for each one separately, and somehow combining these forecasts into an overall forecast. In our analysis, we have relied on the METR dataset to estimate our model; however, this approach actually opens up a wealth of opportunities in terms of estimating progress by ablating existing models (e.g., disabling reasoning, varying the training procedure, etc.). While this strategy may be computationally expensive, it can substantially enlarge the available data, enabling the design of more accurate and reliable forecasts. Finally, scaling laws for LLMs (Hestness et al., 2017; Kaplan et al., 2020; Hoffmann et al., 2022) focus exactly on decomposing LLM performance as a function of training dataset size, model size, and training time; they have also been extended to test-time scaling (Snell et al., 2024). This work could form the basis of more reliable forecasts.

In tandem, Section 6 highlighted a number of limitations inherent to evaluating forecasting methodologies, namely the limited amount of evaluation data and the need for extrapolation. We believe that better evaluation methodologies are critical to improving the rigor of these forecasts. Existing methodologies for evaluating time-series data (Hyndman & Athanasopoulos, 2018) as well as recently-developed online uncertainty quantification algorithms (Gibbs & Candes, 2021; Bastani et al., 2022; Angelopoulos et al., 2024; Noarov et al., 2025; Ge et al., 2025) could play a critical role in solving this problem.

**Venues for academic discussion.** Currently, there are limited venues for rigorous academic discussion about AI forecasting. Given the substantial implications both for policy and for individual decision-making, we argue that forecasting work requires significantly more review. We believe that providing venues for academic discussion of forecasting

work. More broadly, we also call on academic researchers (especially researchers interested in AI safety, the future of work, etc.) to engage with this kind of work, since it is an important piece of understanding the long-term implications of increased AI capabilities. The difference between exponential, linear, and plateauing growth significantly impacts discussions on the most important challenges in both AI safety and societal implications of LLMs.

## 8. Related Work

**Benchmarking LLMs.** A number of complex evaluation frameworks have been proposed to assess AI performance under realistic conditions. SWE-bench evaluates LLMs on real-world GitHub pull requests paired with gold-standard fixes and unit tests (Jimenez et al., 2023). GDPval compiles a comprehensive set of tasks designed to be representative of the U.S. economy (Patwardhan et al., 2025); domain experts from each representative industry are recruited to design tasks for AI systems, while additional experts evaluate model outputs and annotate whether AI-generated solutions are preferred over human completions. Ho et al. (2025) propose a method for stitching together existing benchmarks to assess long-term AI performance. Wijk et al. (2024) introduce RE-bench, which collects challenging research-and-development tasks hand-crafted by human experts.

**Forecasting AI capabilities.** Maslej et al. (2025) systematically documents advances and trends in AI across multiple domains over time, providing an empirical basis for forecasting. Epoch AI has published papers and reports offering in-depth analyses of AI capabilities and future development trajectories (e.g., Owen (2025)). Sinha et al. (2025) distinguish between "execution" and "planning," focusing on a model's ability to correctly execute a complex but predefined plan. They show that diminishing improvements in single-step accuracy can compound, resulting in exponential growth in the length of tasks a model can complete.

## 9. Conclusion

We have argued that the increase in LLM capabilities may not be exponential; instead, it is plateauing or linear. Our forecasting methodology combines domain-specific modeling that decomposes progress into separate base and reasoning capabilities, with an empirical analysis based on the METR dataset, which we believe could form the basis of future forecasting models. Importantly, our methodology is primarily intended to be a compelling alternative rather than a definitive rebuttal. We believe that substantially more work needs to be done both improving forecasting methodologies as well as improving the evaluation of these methodologies, and call for the community to work towards these goals.

## A. Proof of Theorem 4.1

First, suppose that for some $j \in \{0, 1, ..., k-1\}$, we have $x \in [j\alpha, j'\alpha]$, where $j' = j + 1$. For all $i \leq j$,

$$f_i(x) = 1 - \frac{e^{-x+i\alpha}}{1 + e^{-x+i\alpha}} \geq 1 - \frac{e^{-(j-i)\alpha}}{2},$$

so we have

$$\prod_{i=1}^{j} f_i(x) \geq 1 - \frac{1}{2}\sum_{i=1}^{j} e^{-(j-i)\alpha} \geq 1 - \frac{1}{2}\sum_{h=0}^{\infty} e^{-h\alpha}$$

$$\geq 1 - \frac{1}{2}\sum_{h=0}^{\infty}\frac{1}{3^h} \geq \frac{1}{4},$$

since $e^{-2} \leq 1/3$. In addition, we have $f_i(x) \leq 1$, so $\prod_{j=1}^{i} f_i(x) \leq 1$. Next, for all $i \geq j'$, we have

$$f_i(x) = \sigma(x - i\alpha) = \frac{e^{x-i\alpha}}{1 + e^{x-i\alpha}} \geq \frac{e^{-(i-j)\alpha}}{1 + e^{-(i-j')\alpha}},$$

so

$$\prod_{i=j'}^{k} f_i(x) \geq \frac{\prod_{i=j'}^{k} e^{-(i-j)\alpha}}{\prod_{i=j'}^{k}(1 + e^{-(i-j')\alpha})}.$$

For the numerator, we have

$$\prod_{i=j'}^{k} e^{-(i-j)\alpha} = \prod_{h=1}^{k-j} e^{-h\alpha} = \exp\left(-\alpha\sum_{h=1}^{k-j} h\right)$$

$$= \exp\left(-\frac{\alpha}{2}\cdot(k-j+1)(k-j)\right).$$

For the denominator, we have

$$\prod_{i=j'}^{k}(1 + e^{-(i-j')\alpha}) \leq \prod_{h=0}^{\infty}(1 + e^{-h\alpha})$$

$$= \exp\left(\sum_{h=0}^{\infty}\log(1 + e^{-h\alpha})\right) \leq \exp\left(\sum_{h=0}^{\infty} e^{-h\alpha}\right)$$

$$\leq \exp\left(\frac{1}{1 - e^{-\alpha}}\right) \leq 5.$$

In addition, $f_i(x) \leq e^{x-i\alpha} \leq e^{-(i-j')\alpha}$, so

$$\prod_{i=j'}^{k} f_i(x) \leq \exp\left(-\frac{\alpha}{2}\cdot(k-j')(k-j)\right).$$

Putting everything together, we have

$$\frac{1}{20}\exp\left(-\frac{\alpha}{2}\cdot(k-j+1)(k-j)\right)$$

$$\leq f(x) \leq \exp\left(-\frac{\alpha}{2}\cdot(k-j-1)(k-j)\right).$$

Next, if $x \geq k\alpha$, a similar argument shows that

$$\frac{1}{4} \leq f(x) \leq 1.$$

Finally, if $x \leq 0$, then we have

$$\prod_{i=1}^{k} f_i(x) \geq \frac{\prod_{i=1}^{k} e^{x-i\alpha}}{\prod_{i=1}^{k}(1 + e^{x-i\alpha})}.$$

For the numerator, we have

$$\prod_{i=1}^{k} e^{x-i\alpha} = e^{kx}\exp\left(-\alpha\sum_{i=1}^{k} i\right)$$

$$= e^{kx}\exp\left(-\frac{\alpha}{2}\cdot k(k+1)\right),$$

and the denominator is bounded by 5 as before. In addition,

$$\prod_{i=1}^{k} f_i(x) \leq \prod_{i=1}^{k} e^{x-i\alpha} \leq e^{kx}\exp\left(-\alpha\sum_{i=1}^{k} i\right)$$

$$= e^{kx}\exp\left(-\frac{\alpha}{2}\cdot k(k+1)\right).$$

Thus, we have

$$\frac{1}{5}e^{kx}\exp\left(-\frac{\alpha}{2}\cdot k(k+1)\right)$$

$$\leq f(x) \leq e^{kx}\exp\left(-\frac{\alpha}{2}\cdot k(k+1)\right).$$

The claim follows. $\square$

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
