# OpenReview forum: "Position: AI Capabilities May Not Increase Exponentially"
_ICML.cc/2026/Position_Paper_Track — ICML 2026 Position Paper Track regular_

### Official Review · Reviewer_7TnY · 2026-02-27

**Significance:** 3
**Argument Clarity:** 3
**Rating:** 5
**Confidence:** 3

**Questions:**

I would like to thank the authors for this interesting read. I feel that their analysis is carefully motivated and provides very interesting insights about the future of LLM’s capabilities. Their reasoning is sound, and the logical flow feels natural. My questions are mainly related to the weaknesses above (which are more suggestions than the actual weaknesses)

1.	Can we adapt probabilistic forecasting to this case? Specifically, I wonder whether we can do some sort of interdependent sampling for the forecasts to see what the different potential futures may look like (see, for instance, Efficiently Generating Correlated Sample Paths from Multi-step Time Series Foundation Models by Baron et al.).
2.	Is there any correlation between the number of tools used by the models and the observed performance? May it be that such an axis will spur a new exponential growth phase? Same for energy-based and non-autoregressive models mentioned above.
3.	Is it possible to draw lessons from other domains where we can reach the inflection point? For instance, in computer vision, are there any benchmarks that can serve as a backtest for the forecasting model given an arguably richer history of breakthroughs in this field?

**Alternative Views Section:**

Yes

**Compliance With Llm Reviewing Policy A Conservative:**

Affirmed.

**Discussion Potential:**

3

**Final Justification:**

I maintain my already favourable evaluation of this paper.

**Paper Summary:**

This paper puts forward and empirically verifies a position that questions the exponential growth prediction of the AI capabilities suggested by prior work. The authors start by carefully outlining the arguments that prompt them to consider different trends than those used in the MERT study. One of them is that a reasonable candidate for fitting a curve that extrapolates AI capabilities should be done by taking into account a potential inflection point after which such capabilities will plateau. The second aspect is to carefully disentangle the effect of base model capabilities from that of reasoning. Indeed, the benefits of pre-training scaling have long been under scrutiny, suggesting that the models may have reached a plateau in this sense. This makes the multiplicative model that accounts for the two factors in a separate, yet interdependent way, meaningful. Overall, the authors provide a careful and well-motivated prediction for the future of AI models that may prompt an interesting debate within industry and academia.

**Position:**

Yes

**Position In Title:**

Yes

**Related Work:**

3

**Strengths And Weaknesses:**

*Strengths*

1.	Careful analysis of the prior work suggesting exponential growth of AI capabilities
2.	Well-motivated model that accounts for domain-specific factors behind the growth of AI capabilities
3.	Honest evaluation of the model’s limitations and outline for future improvements

*Weaknesses*

1.	Similar to the MERT model, it is regrettable that the proposed study does not provide a probabilistic forecast and uses only pointwise evaluation metrics (MSE). It is indeed a common practice to use probability metrics such as CRPS or WQL for assessing the accuracy of the probabilistic fit.
2.	While authors acknowledge this in their review, it would also be interesting to understand the different scaling axes of reasoning: sequential one (CoT) and parallel one (self-consistency, external verifiers, etc.) as different multiplicative factors of their model.
3.	A study feels somewhat incomplete without a brief discussion of what new emerging technologies are currently in development that may help to avoid the currently predicted inflection points. This may include the use of faster non-autoregressive models (diffusion-based or energy-based).

**Support:**

3

---

> ### Author Rebuttal · Authors · 2026-03-31
>
> **Weakness 1:** We agree with the reviewer that CRPS and WQL can greatly improve the reliability of AI forecasting models. In fact, we believe there is a large set of important tools and results in the existing statistics and machine learning literature that can be beneficial; we list a few of them in the call to action section of our paper. The main goal of our paper aims to draw the broader attention to importance of AI capability forecasting, and to highlight potential directions for future work.
>
> **Weakness 2:** We agree with the reviewer that our current model fails to include many important model characteristics. An ideal specification should include critical components such as model size, attention architecture, training data curation, inference-time methods, and used tools. However, since the frontier models we tested are all closed models, we do not have access to the information, making our model less accurate. We hope to run more comprehensive analyses in the future with more collaboration from the community.
>
> **Weakness 3:** We thank the reviewer for these suggestions. There are many promising new advancements that can boost model capabilities significantly, including diffusion language models, synthetic data generation, and improved attention architecture for long context. Our framework can incorporate the new developments if model providers share relevant details of their models. We will discuss some of these technologies in a revision.
>
> **Q1:** It is definitely possible to generate sample paths as long as we have sufficient data. It is possible to collect significantly more data under our framework, provided that important model characteristics such as size, architecture, and training data are accessible. We can potentially benchmark a large collection of open-source models to expand the datasets. We currently only have 15 data points due to the lack of available characteristics for frontier closed models.
>
> **Q2:** It is possible that the number and the quality of tools constitutes an important source of AI capability growth. In this paper, we treat tools as external components. For example, a coding tool can be compatible with many AI models and the final quality depends on AI's ability to use the given tool. If tools become more complex and powerful, AI models capabilities also need to improve to effectively use the tools. We hope to benchmark models with emerging technologies, such as non-autoregressive models, in future work.
>
> **Q3:** We thank the reviewer for the suggestion! It is indeed a interesting idea to borrow experience from other domains, but it is beyond the scope of our paper, and we leave it to future work.

---

> > ### Author Rebuttal · Reviewer_7TnY · 2026-04-02
> >
> > Thank you for your reply. I maintain my already favourable score.

---

### Official Review · Reviewer_ARDw · 2026-03-13

**Significance:** 3
**Argument Clarity:** 4
**Rating:** 5
**Confidence:** 3

**Questions:**

(1)  Is it possible to say anything conclusively from 15 data points? There are more models.

(2) What is the reason for choosing a multiplicative relationship? The paper states that it provides a growth curve that is qualitatively consistent with METR, but this doesn’t clarify the choice.

**Alternative Views Section:**

Yes

**Compliance With Llm Reviewing Policy A Conservative:**

Affirmed.

**Discussion Potential:**

4

**Final Justification:**

I maintain my positive score, but I highly recommend that "views held by some researchers", this needs to be substantiated with citations. Who are the researchers? Where have they mentioned these view? Please link/cite sources!

**Paper Summary:**

The paper directly argues against the position that “AI capabilities have exhibited exponential growth” stating that plateauing is supported by the same data. The work demonstrates that with the same data points, one can fit a sigmoid curve, rather than the exponential one provided by a prior work METR, 2025., that is to say that neither hypothesis can be ruled out just by fitting the data points to a curve.

**Position:**

Yes

**Position In Title:**

Yes

**Related Work:**

4

**Strengths And Weaknesses:**

Strengths:
(1) The paper expresses uncertainty in the introduction, deeming that it is impossible to rule out exponential or plateauing growth, but makes a case for plateauing also being as plausible as exponential growth. Expressing uncertainty supports the paper’s call for more rigor.

(2) The paper does a good job of refitting a sigmoid with identical data as METR, this makes for a compelling case, not only for AI capabilities not being exponential, but also for performing similar evaluations in the future.

Weaknesses:

(1) The position in the title discusses how AI Capabilities Are Not Increasing Exponentially. But the discussion regarding the alternative views, the paper discusses several opinions (uncited and unsubstantiated) against forecasting AI capabilities as a whole. It would be beneficial to understand why the paper links a discussion on the true rate of change of AI capabilities to “why should we forecast AI capabilities?”

(2) A weakness that is shared from METR to here, the dataset is very limited.

**Support:**

3

---

> ### Author Rebuttal · Authors · 2026-03-31
>
> **Weakness 1:** We apologize for the confusion. We include these opinions in the alternative views sections to reflect views held by some researchers that efforts should not be devoted to identifying a scientific AI forecasting method, which runs counter to the central claim of this paper. We intend to challenge the dominant forecasting models and invite the AI community to engage in scientific discussions on AI capability forecasting.
>
> **Weakness 2:** We agree with the reviewer that the limited dataset is a significant weakness. However, we are only arguing that our model is one plausible model in a large space of potential models. We aim to show that the dominant view proposed by METR is not the only plausible interpretation of the limited data.
>
> **Q1:** It is possible to collect significantly more data under our framework, provided that important model characteristics such as size, architecture, and training data are accessible. We can potentially benchmark a large collection of open-source models to expand the datasets. We currently only have 15 data points due to the lack of available characteristics for frontier closed models. With more granular data, our forecasting methods can become more accurate.
>
> **Q2:** We are positing the multiplicative relationship as one plausible explanation of how AI capabilities improve over time; it is not intended to be the one ``correct'' model, but an illustration of an interesting possibility. As we establish in Theorem 4.1, the product of sigmoid curves grows at an exponential rate and plateaus eventually. The theoretical result explains the exponential-like data patterns and proposes a fundamental constraint on the growth, namely continuous emergence new technologies.

---

> > ### Author Rebuttal · Reviewer_ARDw · 2026-04-02
> >
> > I maintain my positive score, but I highly recommend that "views held by some researchers", this needs to be substantiated with citations. Who are the researchers? Where have they mentioned these view? Please link/cite sources!

---

### Official Review · Reviewer_YKK2 · 2026-03-13

**Significance:** 3
**Argument Clarity:** 3
**Rating:** 5
**Confidence:** 3

**Questions:**

How can we be more confident in the functions/models we fit to the progress curve--are there more fundamental/theoretically justified ways of picking these?

**Alternative Views Section:**

Yes

**Compliance With Llm Reviewing Policy A Conservative:**

Affirmed.

**Discussion Potential:**

4

**Final Justification:**

The overall argument was convincing given what the authors assumed, and is of high impact to the broader community. The authors succesfully responded to my questions/weaknesses, so I am confident the position paper will be of high impact.

**Paper Summary:**

The authors position themselves against the perspective that AI capabilities are increasing exponentially, and primarily argue against a METR study regarding exponential AI capabilities that, when extrapolated, results in AI being able to automate much of what takes humans a month. The authors primarily do this via a new model as well as a philosophical argument regarding the fact that much of recent progress has been driven by a new paradigm (reinforcement learning on verifiable rewards), which looks to bring about exponential progress now but later will taper off.

**Position:**

Yes

**Position In Title:**

Yes

**Related Work:**

2

**Strengths And Weaknesses:**

#### Strengths
- The proposed new curves, particularly Figure 4, are astonishing and really demonstrate the gap between different assumptions (METR vs their sigmoid link) and the vastly different results in what AI can do.
- The analysis of models/functions was thorough
- The paper's positioning and discussion are of extremely high impact, as they have potential for massive economic impact.


#### Weaknesses
- The curves picked for fitting still remain highly speculative, and although the authors did a good job explaining their choices, this makes it harder to believe results (both their results as well as from METR).
- The paper's central position, as noted by the authors, depends on no new major AI breakthroughs being made, which is a relatively strong assumption, and makes the paper's take more nuanced.
- The pape primarily references a METR study, and delves less into other more nuanced takes/studies regarding AI exponential progress
- Perhaps the authors could have proposed a differerent metrics for measuring progress that is more robust/high variance than task duration

**Support:**

2

---

> ### Author Rebuttal · Authors · 2026-03-31
>
> **Weakness 1:** We believe that AI capability forecasting remains a open but crucial discussion. The paper intends to highlight the substantial uncertainty in exponential growth in AI capabilities, especially in the longer term. The goal of our model is to highlight how incorporating mechanisms for driving progress might improve predictions; however, we do not claim it to be the only valid approach.
>
> **Weakness 2:** We propose that the exponential-looking growth curve can be decomposed into a series of plateauing technological advancements. We do not claim that exponential growth is incorrect, or that growth will necessarily plateau in the near future; our main goal is to provide an alternative view on the source of the exponential growth and comment on its implications for future growth. Thus, our central point is more accurately: "continuing exponential growth requires future AI breakthroughs".
>
> **Weakness 3:** We agree with the reviewer that there are other important works on AI capabilities forecasting. We primarily focus on the METR study given its significant impact among both AI researchers and practitioners.
>
> **Weakness 4:** The discussion on identifying the appropriate metric remains an important discussion. We agree that effort should be devoted to identifying better/more robust metrics, but it is beyond the scope of our work and we leave it to future research.
>
> **Q1:** We believe future AI forecasting methods should carefully consider various aspects of model characteristics (e.g., size, training data, architecture) and draw support from existing AI research. Our model challenges the simplistic view that forecasting is solely about curve-fitting in a dataset. As shown in our results, the simple curve-fitting approach is fragile and can be misleading. Our approach of decomposing model capabilities into separate technologies provide a reasonable alternative model for AI forecasting. With more granular data on model characteristics, our approach becomes more accurate. However, we believe that the discussion on fundamental approaches to evaluate AI forecasting methods remains open.

---

> > ### Author Rebuttal · Reviewer_YKK2 · 2026-04-02
> >
> > I feel all of my concerns have been addressed thoroughly with the rebuttal as well as the reviews/rebuttals to other authors. I found the overall argument and positioning that "the exponential-looking growth curve can be decomposed into a series of plateauing technological advancements" to be convincing and robust given what the authors assume.
> >
> > I have increased my score to a 5 consequently.

---

### Official Review · Reviewer_Hech · 2026-03-14

**Significance:** 3
**Argument Clarity:** 3
**Rating:** 5
**Confidence:** 4

**Questions:**

Here, I talk about my questions, comments, and weaknesses.

1. The base LLM's capabilities are rather ill-defined; it already includes multiple components, such as test-time inference. Additionally, components such as agentic capabilities could have been factored into the argument.
2. As already stated in the paper, MSE is used to compare models, and adding other metrics could have made the argument convincing.
3. Multiplicative assumption: It would have been nice to see other assumptions here. While the multiplicative assumption is explanatory, it's hard to argue whether it's most sensible.

**Alternative Views Section:**

Yes

**Compliance With Llm Reviewing Policy A Conservative:**

Affirmed.

**Discussion Potential:**

3

**Paper Summary:**

The paper argues that AI capabilities are not increasing exponentially. Their claim dismisses the METR report that suggested otherwise. The empirical support for this position is well presented, and the theoretical statements also strengthen their argument. Finally, the call to action provides the necessary actions for their ML community.

**Position:**

Yes

**Position In Title:**

Yes

**Related Work:**

3

**Strengths And Weaknesses:**

The paper separates the forecasting part for AI capabilities from its argument, which is sensible given that they provide an alternative explanation for the phenomenon observed about AI capabilities. More importantly, the use of the sigmoid function easily explains the observed phenomenon, something that was not thoroughly explored in the METR analysis. The breakdown of capabilities into advances and plotting separate curves cleverly shows the dependence on new ideas for continuing development of AI.

I provide weaknesses in the questions section.

**Support:**

3

---

> ### Author Rebuttal · Authors · 2026-03-31
>
> **Q1:** We agree with the reviewer that LLM base capabilities are under-specified. An ideal specification should include critical components such as model size, attention architecture, training data curation, inference-time methods, and used tools. However, since the frontier models we tested are all closed, we do not have access to this information, making our model less accurate. Thus, the primary goal of our paper is to highlight the importance of understanding the mechanisms underlying increasing AI capabilities. We hope to run more comprehensive analyses in the future with collaboration from the community.
>
> **Q2:** We agree with the reviewer that our use of in-sample MSE is an imperfect proxy for model comparisons. Nonetheless, it is a useful metric to evaluate performance of forecasting models. We hope to conduct out-of-sample tests in the future when model performance data become more abundant.
>
> **Q3:** The multiplicative assumption is one of many possible  specifications to model the relationship between model release date and model capabilities, and is intended to demonstrate the fragility of the exponential assumption that has been used in the literature. We hope to test a larger selection of assumptions in the future and encourage the community to also scrutinize the assumptions in existing forecasting models.

---

> > ### Author Rebuttal · Reviewer_Hech · 2026-04-04
> >
> > Thanks for the rebuttal. I still think trying out more assumptions could have been nicer, but I agree that this paper showed the fragility of earlier assumptions. In light of this, I keep my positive evaluation.

---

### Decision · Program_Chairs · 2026-04-30

**Decision:**

Accept (regular)

**Comment:**

The debate persists over whether AI advancement is exponential or slowing only linear or even down, with pure scaling of models facing physical and economic constraints. While some data suggests AI capabilities are doubling roughly every seven months, competing analyses suggest that easy gains are diminishing, making continued exponential growth difficult to sustain. This is an important take-away message, and the reviewers agree that the position paper should be accepted. I fully agree.